# Optimization of Medium Constituents for the Production of Citric Acid from Waste Glycerol Using the Central Composite Rotatable Design of Experiments

**DOI:** 10.3390/molecules28073268

**Published:** 2023-04-06

**Authors:** Ewelina Ewa Książek, Małgorzata Janczar-Smuga, Jerzy Jan Pietkiewicz, Ewa Walaszczyk

**Affiliations:** 1Department of Agroengineering and Quality Analysis, Faculty of Production Engineering, Wroclaw University of Economics and Business, Komandorska 118–120, 53-345 Wrocław, Poland; 2Department of Food Technology and Nutrition, Faculty of Production Engineering, Wroclaw University of Economics and Business, Komandorska 118–120, 53-345 Wrocław, Poland; 3Department of Human Nutrition, Faculty of Health and Physical Culture Sciences, Witelon Collegium State University, Sejmowa 5A, 59-220 Legnica, Poland; 4Department of Process Management, Faculty of Production Engineering, Wroclaw University of Economics and Business, Komandorska 118–120, 53-345 Wrocław, Poland

**Keywords:** citric acid, *Aspergillus niger*, design of experiments, response surface methodology, optimization, glycerol

## Abstract

Citric acid is currently produced by submerged fermentation of sucrose with the aid of *Aspergillus niger* mold. Its strains are characterized by a high yield of citric acid biosynthesis and no toxic by-products. Currently, new substrates are sought for production of citric acid by submerged fermentation. Waste materials such as glycerol or pomace could be used as carbon sources in the biosynthesis of citric acid. Due to the complexity of the metabolic state in fungus, there is an obvious need to optimize the important medium constituents to enhance the accumulation of desired product. Potential optimization approach is a statistical method, such as the central composite rotatable design (CCRD). The aim of this study was to increase the yield of citric acid biosynthesis by *Aspergillus niger* PD-66 in media with waste glycerol as the carbon source. A mathematical method was used to optimize the culture medium composition for the biosynthesis of citric acid. In order to maximize the efficiency of the biosynthesis of citric acid the central composite, rotatable design was used. Waste glycerol and ammonium nitrate were identified as significant variables which highly influenced the final concentration of citric acid (Y_1_), volumetric rate of citric acid biosynthesis (Y_2_), and yield of citric acid biosynthesis (Y_3_). These variables were subsequently optimized using a central composite rotatable design. Optimal values of input variables were determined using the method of the utility function. The highest utility value of 0.88 was obtained by the following optimal set of conditions: waste glycerol—114.14 g∙L^−1^and NH_4_NO_3_—2.85 g∙L^−1^.

## 1. Introduction

Citric acid is present in plant tissues as well as animal tissues such as blood, bone or muscle. For living organisms, citric acid is one of the essential carboxylic acids of the Krebs cycle, resulting in the oxidation of glucose to carbon dioxide and water with the release of energy [1,2,3].

Currently, the annual production of citric acid reaches about 1.8 million tons in the world, and the citric acid market is one of the fastest-growing segments of food additives [4,5]. Citric acid has been used in the food, pharmaceutical, chemical and even metallurgical industry due to its harmless nature, chelating and sequestering properties [6]. The reason for the continuous increase in the production of citric acid is its wide range of applications not only in the food and pharmaceutical industry, but also in the production of biopolymers, environment protection and biomedicine [7]. The industrial production of citric acid is dominated by the method of submerged culture using *Aspergillus niger* strains [6,8]. The main substrates in the production of citric acid are beetroot molasses, sucrose, and glucose syrup [2]. Due to the growing demand for citric acid in the world, scientists are conducting research on its production technology to improve the efficiency of the bioprocess and reduce production costs. Currently, research focuses on the use of unconventional raw materials in the biosynthesis of citric acid. The investigated substrates are mainly fruit and vegetable processing waste, refined fatty acids, bran, and crude glycerol [9,10,11,12,13,14,15,16,17].

Glycerol is formed as a by-product in the production of biofuels for diesel engines-biodiesel. The recent increase in biodiesel production generated the problem of management of the glycerol phase. The glycerin phase contains from 30 to 80% glycerol, and for every 100 kg of obtained biodiesel, there is about 11 kg of glycerin phase [18,19,20].

The largest producer of biodiesel is Europe, followed by North America. In 2016, the production of biodiesel in the countries of European Union reached 25 million liters. In 2021, it is estimated that the biodiesel market will reach 41.18 billion dollars in the world [17,21,22].

The increase in biodiesel production has led to the flooding of the market with large amounts of glycerol and search for new methods of its management. However, the use of the glycerin phase is limited because of its contamination with methanol (from 5 to 9%) and alkali. Crude glycerol purification is expensive and the glycerol market is already saturated. Thus, the price of raw glycerol continues to fall and has a direct impact on biodiesel production costs [23]. The solution to this problem may be the use of waste glycerol fraction or partially purified glycerol as a carbon and energy source for the cultivation of microorganisms [24,25,26,27,28,29,30,31,32]. Glycerol as a reducing carbon source can play an important role as a substrate and source of energy for many microorganisms. In addition, it is a precursor to many cell components and a regulator of such cell processes such as metabolic pathways, redox potential and phosphorus management [33]. Bioconversion of glycerol by microbes allows to overcome the drawbacks associated with its chemical conversion, requiring the use of high temperatures, pressures and high costs [34].

The interest in using glycerol as the main substrate for the biosynthesis of citric acid with the participation of *Aspergillus niger* strains is small, despite the large supply of waste on the market. The literature mainly presents the results of research on citric acid biosynthesis, in which glycerol is an additional carbon source in culture media [35,36,37,38]. This is explained by the fact that the substances contained in glycerol have an inhibitory effect on the bioprocess. The next reason is the catabolic repression of carbon in *Aspergillus niger* strains caused by glycerol [39]. Nevertheless, the prospects for using glycerol in the citric acid production with *Aspergillus niger* strains seems to be promising, but this process requires selection of the right strain and optimization of the culture conditions.

The aim of the research was to use the central rotatable plan and utility functions to determine the optimal culture medium composition for the citric acid biosynthesis with *Aspergillus niger* strains using waste glycerol as a main carbon source.

## 2. Results and Discussion

### 2.1. Statistical Analysis

Results included statistical analysis to verify the homogeneity of Brown–Forsyth’a variance test. The test showed homogeneity of variance (*p* > 0.01).

The coefficients of determination (R^2^) of the predicted models of Y_1,_ Y_2_ and Y_3_ were 0.9265, 0.8882 and 0.9266, respectively. The coefficient of determination informs about the variability part of the dependent variable that was explained by the model. The result set indicated that 80% of the variability was included in these mathematical models. The value of the determination coefficient above 0.75 indicates a high coherence between the predicted and experimental values [40]. The significance of each coefficient was determined using the Fisher (*Snedecor)* test and *p*-value. The corresponding variable is usually considered as significant when the absolute F-value becomes larger and the *p*-value becomes smaller [41]. The F-values of Y_1_, Y_2_ and Y_3_ were 1120.76, 0.0078 and 869.91, respectively, while the *p*-values of Y_1_, Y_2_ and Y_3_ were less than 0.01, which indicated that the models were significant.

The ANOVA analysis showed that the input variables X_1_, X_1_^2^, X_2_ and X_2_^2^ in both the linear and quadratic terms of variability were significant for the Y_1_, Y_2_, and Y_3_ models. It can be readily observed that X_1_X_2_ was significant for the model Y_1_. The respective *p*-values (*p*-values > 0.01) of the models Y_1,_ Y_2_ and Y_3_ in *Lack of Fit* test were 0.0312, 0.0202 and 0.0510, which demonstrated that *Lack of Fit* test for the three models was not significant. These results indicated that the models were applicable to accurately predicting the variation. Thus, the mathematical models were satisfactory to perform statistical analyses [42].

Figure 1, Figure 2 and Figure 3 show the response surface plots of changes in the output variables as a function of the crude glycerol concentration (X_1_) and the ammonium nitrate concentration (X_2_). The response surface analysis was conducted with the following constant concentrations: KH_2_PO_4_ = 0.2 g·L^−1^ and MgSO_4_∙7H_2_O = 0.2 g·L^−1^. The highest concentration of citric acid (Y_1_ = 38.15 g·L^−1^) was achieved when the initial concentration of waste glycerol and ammonium nitrate was 120.0 g·L^−1^ and 3.0 g·L^−1^, respectively. In the case of decrease in ammonium nitrate concentration of 2.0 g·L^−1^, a significant decrease in the final concentration of citric acid (Y_1_ = 11.32 g·L^−1^) was observed. The lower concentration of crude glycerol (80.0 g·L^−1^) and ammonium nitrate (1.8 g·L^−1^) in the culture medium resulted in noticeable reduction in final citric acid concentration (Y_1_ = 5.60 g·L^−1^) (Figure 1). In addition, the highest volumetric rate of citric acid biosynthesis (Y_2_ = 0.106 g·L^−1^∙h^−1^) was obtained at the initial concentration of crude glycerol of X_1_ = 120 g·L^−1^ and ammonium nitrate X_2_ = 3.0 g·L^−1^ in the culture medium. Similarly, reduction in the nitrogen source to X_2_ = 2.0 g·L^−1^ resulted in a significant reduction in the volume rate of citric acid biosynthesis (Figure 2). Interaction analysis for the yield of citric acid biosynthesis showed that range of input variables values were X_1_ = 100.0–120.0 g·L^−1^ and X_2_ = 2.5–3.0 g·L^−1^. If concentrations of crude glycerol and ammonium nitrate were reduced below 80 g·L^−1^ and 2.5 g·L^−1^, respectively, significant reduction in the yield of citric acid biosynthesis was observed (Figure 3).

### 2.2. Determination of Optimum Medium Constituents

In order to determine the optimal values of the input variables (X_1_, X_2_ and X_3_) on the grounds of three selected final responses, the final concentration of citric acid (Y_1_), the volumetric rate (Y_2_), and biosynthesis yield (Y_3_), the method of utility function was used [43,44,45,46]. The utility function was defined as the following Equation (1):U = f (u_1_, u_2_, …, u_n_), (1)
and a function that satisfies the following conditions:U′(x) > 0, U″(x) < 0.(2)

The value of each response was evaluated using a dimensionless linear function with values in the interval [0.1]. The detailed u_i_ was assigned as follows: u_i_ = 0 if response was of low value, u_i_ = 0.5 if response was of minimal value, and u_i_ = 1 if response was of high value. After determining the utility function for each output variable, a model of total utility was built using the spline function method for fitting the response surface to the utility value. To obtain the optimal values of the input variables, the optimum method of grid node was used.

Figure 4 shows the profiles of approximate values and their utility determined using the optimum method of grid node. The highest utility value of 0.84 was obtained for the following optimal values of the analysed culture medium composition parameters: waste glycerol concentration—114.14 g·L^−1^ and ammonium nitrate—2.85 g·L^−1^. On the grounds of basic research, constant concentrations of KH_2_PO_4_ = 0.2 g·L^−1^and MgSO_4_∙7H_2_O = 0.2 g·L^−1^ were assumed. For such composition of the culture medium, a triplicated verification experiment was performed. The model was verified and validated by comparing experimental and predicted response values as shown in Table 1. The experimental and predicted response values were consistent with each other, indicating the success of the mathematical model in obtaining optimum experimental conditions.

### 2.3. Biosynthesis of the Citric Acid

The biosynthesis of citric acid was performed on an enlarged scale in the Biomer 10 bioreactor. The final concentration of citric acid (Y_1_), the volumetric rate (Y_2_), the yield of citric acid biosynthesis (Y_3_), and other parameters characterizing the biosynthesis process in submerged cultures in the bioreactor are presented in Table 2. The final concentration of citric acid in the culture medium was significantly higher than that in the verification experiment and amounted to Y_1_ = 69.70 g∙L^−1^. In addition, the bioprocess was characterized by a high yield (61.00%) and the volumetric rate of citric acid biosynthesis (0.183 g∙L^−1^∙h^−1^). Qualitative and quantitative analysis of organic acids showed the presence of only citric acid in the culture medium, which proved the high chemical purity of the obtained product.

The literature lacks studies on the assessment of the impact of waste glycerol on the biosynthesis of citric acid. There are also few comparative studies on the biosynthesis of metabolites by microorganisms in media containing waste or anhydrous glycerol as the main carbon source. This is probably due to the belief that glycerol slows down the growth rate of filamentous fungi and does not favor the production of citric acid by *Aspergillus niger*.

According to research conducted by Wittwen et al. [47], glycerol in *Aspergillus niger* cells is phosphorylated to 3-phosphoglycerol, similar to *Aspergillus nidulans* and *Saccharomyces cerevisiae*. This reaction is catalyzed by glycerol kinase, which in *Saccharomyces cerevisiae* is a product of the GUT1 gene. Then, 3-phosphoglycerol is oxidized to phosphodihydroxyacetone by the FAD+-dependent 3-phosphoglycerol dehydrogenase. Finally, phosphodihydroxyacetone is incorporated into the glycolytic pathway [34,37,48].

Various carbon sources were used for the production of citric acid by *Aspergillus niger*. The highest yields are obtained when monosaccharides (glucose) and disaccharides (sucrose) are used as substrates in submerged cultures. Unfortunately, they increase production costs due to their high price [22]. In order to reduce the cost of citric acid production, cheap carbon sources are sought that can be used by *Aspergillus niger* (e.g., apple pomace, sweet potato hydrolysates, sugar cane cake, pineapple pomace) [49,50,51].

Comparing the results obtained in this study with those of other authors, Zohu et al., in a medium with maize waste, obtained an accumulation of citric acid at the level of 100.4 g∙L^−1^ and a yield of 94.11% in submerged cultures of *Aspergillus niger* SIIM M288 [52]. On the other hand, Hu et al., in a study with the same strain, obtained a yield of 74.9% [53]. In submerged cultures of *Aspergillus niger* using two types of dates, GHARS and MECH DEGALA, as a carbon source, the concentration of citric acid was 42.25 g∙L^−1^ and 36.60 g∙L^−1^, respectively [54]. In a study with beet molasses as a substrate, citric acid concentrations of 19.13 g∙L^−1^ and 34.62 g∙L^−1^ were obtained with a sugar concentration of 200 g∙L^−1^ and 150 g∙L^−1^, respectively [55]. In the process of optimizing the biosynthesis process, Aboyeji et al. obtained a yield of 4.36 mg∙mL^−1^ from sweet potato starch hydrolysates [56].

## 3. Materials and Methods

### 3.1. Major Substrate and Microorganisms

The major substrate, waste glycerol, was collected from Wratislawia Biodiesel S.A. Wrocław, Poland and stored in room at 20 °C. The chemical characteristics of the substrate are presented in Table 3.

The fungal strain of *Aspergillus niger* PD-66 used in this work was obtained from Pure Cultures Collection maintained at the Department of Food Biotechnology and Analysis at the Wroclaw University of Economics and Business. The culture was regularly sub-cultured and maintained at 4 °C. *Aspergillus niger* spores were produced on potato dextrose agar (PDA) plates for 10 days at 30 °C and washed with 25 mL sterilized distilled water to prepare the inoculum. Spore suspension was collected in a 500 mL Erlenmeyer flask.

### 3.2. Experimental Procedure

The culture medium used in the study contained waste glycerol, NH_4_NO_3_ (p.a., Chempur, Piekary Śląskie, Poland) KH_2_PO_4_ (p.a., Chempur, Piekary Śląskie, Poland), MgSO_4_∙7H_2_O (p.a., Chempur, Piekary Śląskie, Poland) and tap water. The initial pH of medium was adjusted to 3.0 by adding 5M HCl. The medium with nutrients was autoclaved at 121 °C for 30 min. According to the created experiment plan, variants of the composition of culture media were obtained, which are presented in Table 4.

For the optimisation procedure, small-scale batch fermentation experiment was conducted in 500 mL Erlenmeyer flasks holding 100 mL of the medium. After autoclaving, the medium was inoculated with a spore suspension of 1 × 10^5∙^mL^−1^. The Erlenmeyer flasks with their contents were incubated in a shaker GFL 3033 (Lauda Dr. R. Wobser GMBH & CO. KG, Lauda-Königshofen Germany) at 30 °C on 200 rpm∙min^−1^ for 15 days. All of above experiments were conducted three times.

The batch cultivations on large scale were performed in BIOMER 10 (model, manufacturer, Wrocław, Poland) bioreactor with total volume of 7 L and working volume of 5 L. Temperature was maintained at 30 °C. To prevent foam formation, rapeseed oil was added automatically using level sensor control. The aeration rate was set 1.0 L∙min^−1^ and the dissolved oxygen tension was kept at 80% saturation. In the subsequent days of the bioprocess, the rotational speed of the agitator shaft was gradually increased from 300 to 800 rpm∙min^−1^.

### 3.3. Determination of Citric Acid and Glycerol

For HPLC analysis, centrifuged samples (8000× *g*, 15 min) were filtered through nylon filters (diameter 0.22 μm) prior to diluting 1:1 with distilled water. For analysis of glycerol and organic acid, Perkin Elmer HPLC (Series 200, Waltham, 940 Winter St, United States model, manufacturer, city, state abbreviation,) was used with the Eurokat H65 (Knauer Wissenschaftliche Geräte GmbH, Berlin, Germany) column at 60 °C, with constant flow rate 0.6 mL∙min^−1^ HPLC water eluent. For detection, RI Perkin Elmer Series 2000 detector and a variable wavelength UV/VIS CE detector (Series 200, Waltham, 940 Winter St, United States) (model, manufacturer, city, (state abbreviation if USA and Canada), country) at 210 nm were used.

### 3.4. Experimental Design

In the present study, Statistica 13.3 software (Statsoft Inc., Tulsa, OK, USA) was employed to perform the experiment design and development results. Analysis of variance (ANOVA) and RSM (Response Surface Methodology) were employed to determine the regression coefficients, statistical significance of the model terms and fit of the experimental data to mathematical models, which aims at optimizing the overall region for three response variables.

A quadratic model used to predict the response variables is shown as the following Equation (3).
Y = *b*_0_ + *b*_1_X_1_ + *b*_2_X_1_^2^ + *b*_3_X_2_ + *b*_4_X_2_^2^ + *b*_5_X_1_X_2,_(3)
where Y is the predicted dependent variable, X_1_ and X_2_ are the independent variables and β_i_ are the regression coefficients.

In the present study, parameters of culture medium for citric acid biosynthesis were optimized using response surface methodology. RSM is a statistical technique used to design experiments, build models, evaluate the effects of the factor and search for the optimal conditions of factors required for desired responses. Moreover, the central composite rotatable design is the most widely used method in RSM. In this study, the central composite rotatable design was used to identify relationship between the response function and bioprocess variables. It was also used to optimize the culture medium parameters of citric acid concentration, rate, and yield of biosynthesis.

Our preliminary experiment demonstrated that the concentration of crude glycerol and ammonium nitrate had significant influence on the final concentration of citric acid, volumetric rate, and yield of its biosynthesis. The values of other bioprocess variables were determined based on previous experiments: KH_2_PO_4_—0.2 g∙L^−1^, MgSO_4_∙7H_2_O—0.2 g∙L^−1^, temperature 30 °C, mixing speed 200 rpm·min^−1^, and pH 3.0. Therefore, the crude glycerol (X_1_) and ammonium nitrate (X_2_) were chosen as independent variables, while response variables were the final concentration of citric acid (Y_1_), volumetric rate (Y_2_) and yield (Y_3_) of citric acid biosynthesis. Then, the natural values of the central point (plan nucleus) and variable step (ΔX_i_) were determined. The natural and coded levels of variables of the central point and variable step are provided in Table 5 and Table 6.

From the variables selected for testing, the rotatable plan was created. The experimental design consists of four factorial points, four axial points at distance 1.414 from the centre and five replicates of the central point. The number of experiments at the central point is 5, which results from the requirement that the variance of the Y value has to be the same at the central point of the plan as well as at the points lying on the sphere within the radius of 1. The experiments were carried out in three independent repetitions in random order. The distance of 1.414 between axial point and central point was calculated from the following Equation (4):(4)$$\alpha =\sqrt[{4}]{2^{n}},$$
where *α* is the distance, *n* is a number of independent variables. In this study, *n* = 2 and *α* = 1.414.

The conformity of the model was determined by regression analysis and ANOVA analysis (*p* < 0.01). The significance of the regression coefficient was analysed by F-test. The relationship between independent variables (X_1_, X_2_) and response variables (Y_1_, Y_2_ and Y_3_) was illustrated by response surface plots.

Table 7 summarizes the experimental and predicted values of the response variables using the central composite rotatable design. ANOVA analysis of response surface quadratic model was used to estimate the relationship between response variables and independent variables of regression models. The results are presented in Table 8, Table 9 and Table 10.

The quadratic models of Y_1_, Y_2_ and Y_3_ in actual and coded variables were demonstrated as the following Equations (5)–(10).

The equation in terms of actual variables:Y_1_ (g·L^−1^) = −210.91 + 70.22X_1_ − 21.02X_1_^2^ + 2.42X_2_− 0.02X_2_^2^ + 0.47X_1_X_2_,(5)
Y_2_ (g∙L^−1^·h^−1^) = −0.69 + 0.25X_1_− 0.06X_1_^2^ + 0.008X_2_ − 0.00005X_2_^2^ + 0.0008X_1_X_2_,(6)
Y_3_ (%) = −250.78 + 2.94X_1_ − 0.02X_1_^2^ + 88.05X_2_ − 21.47X_2_^2^ + 0.31X_1_X_2_.(7)

The equation in terms of coded variables:Y_1_ (g·L^−1^) = 28.30 + 6.17X_1_ − 6.57 X_1_^2^ + 6.03X_2_ − 5.26X_2_^2^ + 4.70X_1_X_2_,(8)
Y_2_ (g∙L^−1^·h^−1^) = 0.08 + 0.01 X_1_ − 0.02X_1_^2^ + 0.02 X_2_ − 0.01 X_2_^2^ + 0.01X_1_X_2_,(9)
Y_3_ (%) = 28.30 + 3.17X_1_ − 7.09X_1_^2^ + 5.72 X_2_ − 5.37X_2_^2^ + 3.07X_1_X_2_.(10)

## 4. Conclusions

The test results confirmed the usefulness of the method of optimizing the culture medium composition for the citric acid biosynthesis using the utility function. Optimization studies included: ANOVA analysis, determination of optimal values of input variables and their interactions as well as mathematical models in the form of quadratic functions describing the dependence of the final concentration of citric acid (Y_1_), volumetric rate (Y_2_), and yield of biosynthesis (Y_3_) on the method parameters. The correlation coefficients of R^2^ for the three models Y_1_, Y_2_, Y_3_ amounting to 92.65%, 88.82%, and 92.65%, respectively, showed good agreement between the experimental and predicted parameters of citric acid biosynthesis. The optimal values of input variables were determined: X_1_ = 114.14 g∙L^−1^ and X_2_ = 2.85 g∙L^−1^. The bioreactor experiment showed that the average final concentration of citric acid was 69.70 g∙L^−1^ and the efficiency coefficient of citric acid biosynthesis was 27.08%∙g∙L^−1^∙h^−1^.

The obtained results confirmed the usefulness of the utility function method to optimize the parameters of biological processes. The presented approach to optimizing the composition of the culture medium using the utility function allows to increase the knowledge about the studied process. Analysis of the results showed that the designed experimental systems allowed identification of relationships describing the interaction of independent variables with response variables. In the studied area of variability, quadratic functions were a satisfactory reflection of the actual bioprocess function. Utility function has enabled poly-optimization by considering three criteria for maximizing the citric acid biosynthesis. The utility function also reflects the purpose of the designed study. The applied utility function determined the feasibility area in which the productivity of the process is maximum. The identified condition allowed to increase the efficiency of the process by 55%.

It should be emphasized that the proposed method makes it possible to determine the optimum based on a number of criteria, the list of which can be freely extended, which is important from a practical point of view.

In addition, this present study specified that crude glycerol by-product from biodiesel production could be used for producing high amount of citric acid without any pretreatment. It is forecast that the production of biodiesel in the years 2023–2025 will amount to 46 billion L, and for every 100 kg of biodiesel obtained, there is approximately 11 kg of glycerin phase [57]. On the other hand, the citric acid market is growing year by year and will reach 3.29 million t by 2028 [58]. This arrangement will enable the use of crude glycerol without pretreatment as a cheap energy source for the biosynthesis of citric acid. This would solve the ecological problems related to the disposal of the glycerine phase generated in the production of biodiesel. The bioconversion of crude glycerol by microorganisms allows to avoid the inconveniences associated with its chemical conversion, involving the use of high temperatures or pressures and consequently high costs [59]. The production of citric acid at a lower cost is in high demand. Therefore, innovations are needed to solve the problems associated with increasing the scale of production in fermentation processes. Hence, cheap substrates, such as by-products from the agro-industry, have the potential to reduce costs and environmental problems.

## Figures and Tables

**Figure 1 molecules-28-03268-f001:**
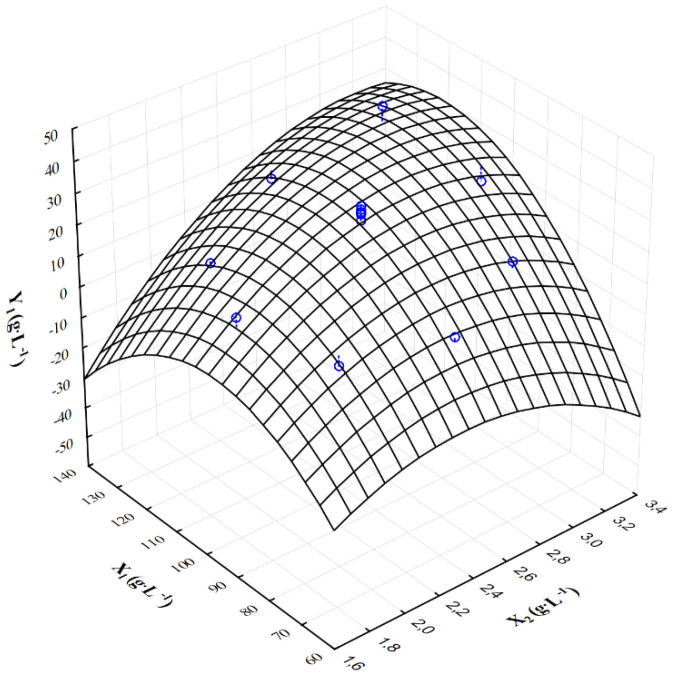
Response surface plot (3D) of the influence of the crude glycerol concentration (X_1_) and ammonium nitrate concentration (X_2_) on the final citric acid concentration (Y_1_); blue points—points where the measurement was made.

**Figure 2 molecules-28-03268-f002:**
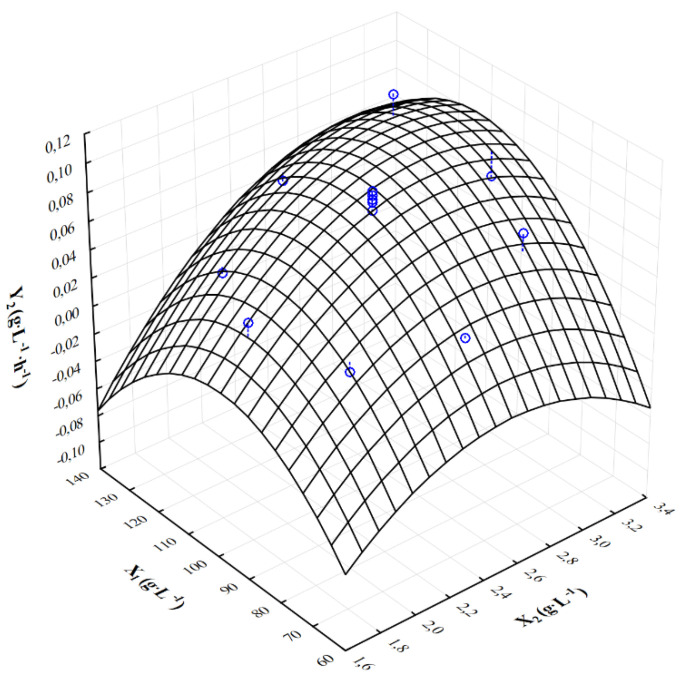
Response surface plot (3D) of the influence of the crude glycerol concentration (X_1_) and ammonium nitrate concentration (X_2_) on the volumetric rate yield of citric acid biosynthesis (Y_2_); blue points—points where the measurement was made.

**Figure 3 molecules-28-03268-f003:**
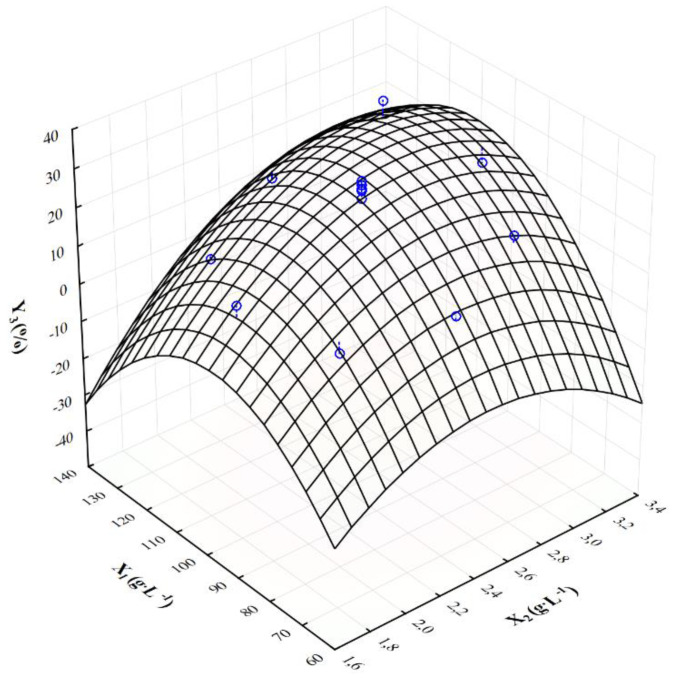
Response surface plot (3D) of the influence of the crude glycerol concentration (X_1_) and ammonium nitrate concentration (X_2_) on the yield of citric acid biosynthesis (Y_3_); blue points—points where the measurement was made.

**Figure 4 molecules-28-03268-f004:**
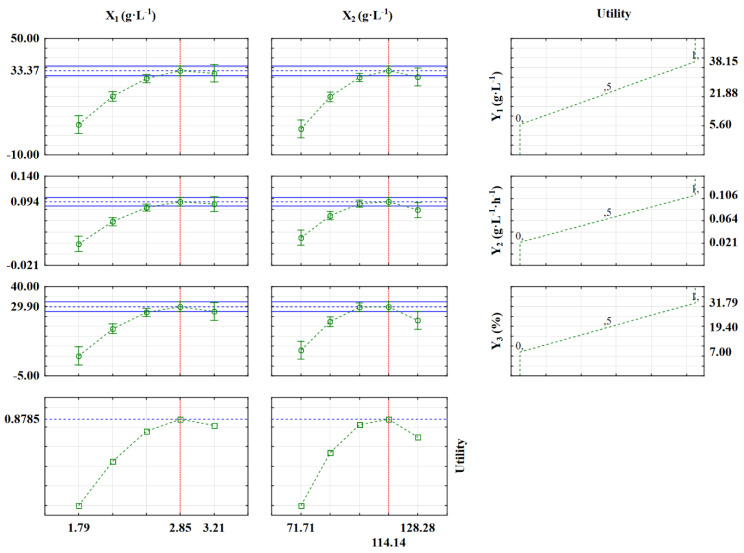
Optimal values utility of the analysed culture medium parameters for the highest final concentration of citric acid (Y_1_), volumetric rate of citric acid biosynthesis (Y_2_), and yield of citric acid biosynthesis (Y_3_). The blue line represents usability. The red line determines the optimal concentrations of the components of the culture medium, waste glycerol and ammonium nitrate.

**Table 1 molecules-28-03268-t001:** Comparison of predicted and experimental response values of verification model.

Response	Optimal Variable Value (g∙L^−1^)		Predicted	Experimental
	Crude Glycerol	Ammonium Nitrate		
Final concentration of citric acid (Y_1_) (g∙L^−1^)	114.14	2.85	36.42	33.69
Volumetric rate of citric acid biosynthesis (Y_2_) (g∙L^−1^∙h^−1^)			0.101	0.095
Yield of citric acid biosynthesis (Y_3_) (%)			32.18	30.26

**Table 2 molecules-28-03268-t002:** Kinetic parameters of citric acid biosynthesis by *Aspergillus niger* PD–66 in optimized medium containing crude glycerol as a carbon source.

Symbol	Unit	Parameters	Results
t	h	Culture time	157
G_K_	g∙L^−1^	Final concentration of glycerol in the medium	0.50
Y_1_	g∙L^−1^	Monohydrate citric acid concentration in culture medium	69.70
Y_2_	g∙L^−1^∙h^−1^	Volumetric rate of monohydrate citric acid biosynthesis	0.183
Y_3_	% (m/m)	Yield of citric acid biosynthesis with respect to introduced substrate	61.00
K_EF_	% g∙L^−1^∙h^−1^	Efficiency coefficient of monohydrate citric acid biosynthesis	27.08
X_K_	g∙L^−1^	Biomass concentration in culture medium	19.40
Y_X/S_	% (m/m)	Yield of biomass biosynthesis	17.00

**Table 3 molecules-28-03268-t003:** The chemical composition of waste glycerol.

Property	Unit	Value
Glycerin	% (m/m)	88.30
Sodium chloride	% (m/m)	3.92
Methanol	% (m/m)	0.04
Water	% (m/m)	6.67
Matter Organic Non-Glycerol	% (m/m)	1.20
pH	-	7.40

**Table 4 molecules-28-03268-t004:** Composition of culture media used to optimize the concentration of waste glycerol and ammonium nitrate.

Run	Crude Glycerol	NH_4_NO_3_	KH_2_PO_4_	MgSO_4_∙7H_2_O
	[g∙L^−1^]	[g∙L^−1^]	[g∙L^−1^]	[g∙L^−1^]
1	80.00	2.00	0.20	0.20
2	80.00	3.00	0.20	0.20
3	120.00	2.00	0.20	0.20
4	120.00	3.00	0.20	0.20
5	71.71	2.50	0.20	0.20
6	128.28	2.50	0.20	0.20
7	100.00	1.79	0.20	0.20
8	100.00	3.20	0.20	0.20
9	100.00	2.50	0.20	0.20
10	100.00	2.50	0.20	0.20
11	100.00	2.50	0.20	0.20
12	100.00	2.50	0.20	0.20
13	100.00	2.50	0.20	0.20

**Table 5 molecules-28-03268-t005:** Actual value of independent variable at the center point and the change in variable step.

Independent Variables, X_i_	Unit	Center Point $X_{i}^{0}=\frac{X_{i}^{\max}+X_{i}^{\min}}{2}$	Variable Step $\Delta X=\frac{X_{i}^{\max}-X_{i}^{\min}}{2}$
X_1_	g∙L^−1^	120.0	20.0
X_2_	g∙L^−1^	2.5	0.5

X—encoded value, X_i_—actual value,
$X_{i}^{0}$—actual value in the center of the domain,
ΔX—variable step.

**Table 6 molecules-28-03268-t006:** Experimental range and levels of variables.

Independent Variables, X_i_	Unit	Levels and Ranges				
		(−1.414)	(−1)	(0)	(+1)	(+1.414)
X_1_	g∙L^−1^	71.71	80.00	100.00	120.00	128.28
X_2_	g∙L^−1^	1.79	2.00	2.50	3.20	3.00

Levels of each variable are axial (±1), central (0) and corner (±1.414).

**Table 7 molecules-28-03268-t007:** Actual and coded variables with the experimental and predicted values.

Run	Actual and Coded Variables				Results					
					Y_1_ (g·L^−1^)		Y_2_ (g∙L^−1^·h^−1^)		Y_3_ (%)	
	Crude Glycerol X_1_		Ammonium Nitrate X_2_		Experimental	Predicted	Experimental	Predicted	Experimental	Predicted
1	80.00	−1	2.00	−1	5.60	8.97	0.021	0.028	7.00	10.02
2	80.00	−1	3.00	1	13.65	11.64	0.063	0.051	17.06	15.32
3	120.00	1	2.00	−1	11.32	11.92	0.031	0.036	9.43	10.21
4	120.00	1	3.00	1	38.15	33.38	0.106	0.091	31.79	27.82
5	71.72	−1.414	2.50	0	7.70	6.44	0.029	0.031	10.74	9.63
6	128.28	1.414	2.50	0	21.23	23.89	0.059	0.065	16.55	18.61
7	100.00	0	1.79	−1.414	12.37	9.26	0.037	0.027	12.37	9.472
8	100.00	0	3.20	1.414	21.82	26.32	0.065	0.082	21.82	25.66
9	100.00	0	2.50	0	28.35	28.30	0.084	0.084	28.35	28.30
10	100.00	0	2.50	0	25.78	28.30	0.077	0.084	25.78	28.30
11	100.00	0	2.50	0	29.28	28.30	0.087	0.084	29.28	28.30
12	100.00	0	2.50	0	30.33	28.30	0.090	0.084	30.33	28.30
13	100.00	0	2.50	0	27.77	28.30	0.083	0.084	27.77	28.30

**Table 8 molecules-28-03268-t008:** Variance analysis of regression equation Y_1._

Source	Sum of Square	Degree of Freedom	Mean Squares	F-Value	*p*-Values
Model	1120.76	5	224.15	17.66	0.0008
X_1_	304.450	1	304.50	103.91	0.0005
X_1_^2^	300.13	1	300.13	102.42	0.0005
X_2_	290.98	1	290.98	99.29	0.0006
X_2_^2^	192.17	1	192.17	65.58	0.0013
X_1_X_2_	88.20	1	88.20	30.10	0.0054
Lack of fit	77.14	3	25.71	8.77	0.0312
Pure error	11.72	4	2.93		
Cor total	1209.62	12			

R^2^ = 92.65%; Adjusted R^2^ = 87.41%; *p *< 0.01 is significant; L—linear term; Q—quadratic term.

**Table 9 molecules-28-03268-t009:** Variance analysis of regression equation Y_2._

Source	Sum of Square	Degree of Freedom	Mean Squares	F-Value	*p*-Values
Model	0.0078	5	0.0016	11.12	0.0032
X_1_	0.0011	1	0.0011	43.61	0.0027
X_1_^2^	0.0023	1	0.0023	87.66	0.0007
X_2_	0.0031	1	0.0031	117.63	0.0004
X_2_^2^	0.0015	1	0.0015	57.82	0.0016
X_1_X_2_	0.0003	1	0.0003	10.21	0.0331
Lack of fit	0.0009	3	0.00029	11.28	0.0202
Pure error	0.0001	4	0.00003		
Cor total	0.0088	12			

R^2^ = 88.82%; Adjusted R^2^ = 80.83%; *p* < 0.01 is significant; L—linear term; Q—quadratic term.

**Table 10 molecules-28-03268-t010:** Variance analysis of regression equation Y_3._

Source	Sum of Square	Degree of Freedom	Mean Squares	F-Value	*p*-Values
Model	869.91	5	9.85	17.67	0.0008
X_1_	80.5393	1	80.5393	27.48	0.0063
X_1_^2^	349.8102	1	349.8102	119.37	0.0004
X_2_	262.0668	1	262.0668	89.43	0.0007
X_2_^2^	200.4206	1	200.4206	68.39	0.0012
X_1_X_2_	37.8140	1	37.8140	12.90	0.0230
Lack of fit	57.2059	3	19.0686	6.51	0.0510
Pure error	11.7219	4	2.9305		
Cor total	938.8329	12			

R^2^= 92.66%; Adjusted R^2^ = 87.41%; *p* < 0.01 is significant; L—linear term; Q—quadratic term.

## Data Availability

The data presented in this study are available on request from the corresponding author.

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
