# Peer review of "Optimization of Medium Constituents for the Production of Citric Acid from Waste Glycerol Using the Central Composite Rotatable Design of Experiments"

_molecules, 2023, doi:10.3390/molecules28073268_

Round 1

Reviewer 1 Report

Dear authors,

The manuscript is prepared according to journal MDPI-Molecules. Aim of the research is in the scope of the mentioned journal. Article presents interesting and current topic of the production of citric acid from waste glycerol.  

1.      The abstract provides aims and important findings of the research.

2.      Introduction: Introduction provides sufficient background and includes all relevant references. The aim and the goal of the research are clearly presented. References should be corrected: line 43: write: [1-3], the same at line 63. Line 47: reference 6 is not written correctly; the same is at line 84.

3.      The conclusion: is clearly presented.

After the minor corrections, the manuscript can be published.

Author Response

Dear Reviewer,

Thank you for your comments on our article reported in the review and good rating.

In lines 44, 48, 64, 85 and 229 the citation has been corrected.

Regards

Authors

Reviewer 2 Report

In this work, the authors investigate the optimization of medium constituents for the production of citric acid from waste glycerol using the central composite rotatable design of experiments. The topic is interesting from a scientific point of view. However, there are some issues that require the authors’ attention.

Lines 106-107: Please add the exact quantities (%) used for the composition of the culture medium.

Line 114: Please add the city and country of origin of the bioreactor’s manufacturer. The same applies for all equipment used in this study.

High repetition rate was discovered in materials and methods (especially 2.4). Try to edit and use appropriate references in each section.

Results/Discussion:

At its current form, mostly results are presented and discussion is limited to the statistical analysis subsection. In my opinion, since the target is citric acid production, the authors should expand their discussion on this matter. The production of citric acid (after the optimization process) and the bioprocess yield rates should be compared to the production of citric acid from other sources. Try answering how the model proposed here leads to a sustainable outcome in terms of efficiency (versus known procedures). In this way, the effectiveness of the model proposed will be more clear to the reader. Lastly, the authors should highlight the use of waste-glycerol and all the advantages derived (environmental, cost-effectiveness, circular economy, etc).

Conclusions:

In my opinion the conclusions are rather technical. It should be edited to incorporate the final outcome (effective citric acid production). Please see my comment above.

Author Response

Dear Reviewer,

Thank you for your comments on our article reported in the review. The article has tried to take into account all the reviewer's comments. I hope that all corrections will improve the quality of the submitted article.

According to the comment in the review, the following changes have been made to the article:

  1. In lines 110 -115, a table with the concentration of culture medium components used in the optimization process has been added.
  2. In lines 119, 121, 130, and 131 the names of the countries from which the equipment used in the research comes were added.
  3. In lines 136, 137, and 147, the repetitions that occurred were corrected.
  4. In line 264, references describing the use of the utility function in research have been added.
  5. In lines 308 - 337, considerations were extended to include a comparison of own results with those of other authors in other substrates. The small amount of literature on the use of waste glycerol for the production of citric acid is also highlighted.
  6. In lines 351-365, information on the benefits of using the research model in maximizing the parameters of the citric acid biosynthesis process from waste glycerol has been added.
  7. Rows 369 - 382 indicate the benefits of using waste glycerol for the production of citric acid.
  8. The article was checked again by a translator specializing in scientific articles http://www.richard.ashcroft.skylan.pl/.

Comment 1. The production of citric acid (after the optimization process) and the bioprocess yield rates should be compared to the production of citric acid from other sources

The considerations in the article were extended with information on the metabolism of glycerol by Aspergillus niger, and comparing it to other sources of carbon from the agro-industry, such as fruit pomace or starch hydrolysates.

Comment 2. How the model proposed here leads to a sustainable outcome in terms of efficiency (versus known procedures). In this way, the effectiveness of the model proposed will be more clear to the reader. 

The article was extended by literature, which describes the application of the method mentioned in this article. It was emphasized that the utility function is used in the description of biological processes and identifies the area in which we obtain the highest process productivity. The parameters obtained in the model allowed for a significant increase in the efficiency of the process on a larger scale.

Comment 3. The authors should highlight the use of waste-glycerol and all the advantages derived (environmental, cost-effectiveness, circular economy, etc).

The article was extended with information on the forecasts of glycerol and citric acid production based on the OECD report. It was noted that the use of microorganisms to process the waste fraction from biodiesel production would eliminate the high costs and environmental problems of waste glycerol processing.

Thank you for all your valuable comments on our article.

Regards
Authors

Round 2

Reviewer 2 Report

The authors have addressed my comments